# Self-assembly of 1T/1H superlattices in transition metal dichalcogenides

Chaojie Luo [1,2,3,6], Guohua Cao[1,2,3,6], Beilin Wang [1,2,3,6], Lili Jiang[1,2,3], Hengyi Zhao[1,2,3], Tongrui Li [4], Xiaolin Tai[5], Zhiyong Lin[1,2,3], Yue Lin [1], Zhe Sun [4], Ping Cui [1,2,3] ✉, Hui Zhang [1,2,3] ✉, Zhenyu Zhang [1,2,3] & Changgan Zeng [1,2,3] ✉

Heterostructures and superlattices composed of layered transition metal dichalcogenides (TMDs), celebrated for their superior emergent properties over individual components, offer significant promise for the development of multifunctional electronic devices. However, conventional fabrication techniques for these structures depend on layer-by-layer artificial construction and are hindered by their complexity and inefficiency. Herein, we introduce a universal strategy for the automated synthesis of TMD superlattice single crystals through self-assembly, exemplified by the $NbSe_{2-x}Te_x$ 1T/1H superlattice. The core principle of this strategy is to balance the formation energies of T (octahedral) and H (trigonal prismatic) phases. By adjusting the Te to Se stoichiometric ratio in $NbSe_{2-x}Te_x$, we reduce the formation energy disparity between the T and H phases, enabling the self-assembly of 1T and 1H layers into a 1T/1H superlattice. The resulting 1T/1H superlattices retain electronic characteristics of both 1T and 1H layers. We further validate the universality of this strategy by achieving 1T/1H superlattices through substituting Nb atoms in $NbSe_2$ with V or Ti atoms. This self-assembly for superlattice crystal synthesis approach could extend to other layered materials, opening new avenues for efficient fabrication and broad applications of superlattices.

Layered transition metal dichalcogenides (TMDs) have attracted considerable attentions in recent years for their rich physical properties[1-10]. The heterostructures and superlattices of TMDs, meticulously assembled from two-dimensional (2D) TMD layers aligned along the c-axis, exhibit the unique capability to manifest the inherent characteristics of their constituent materials. For example, the $WTe_2$/$NbSe_2$ heterostructure simultaneously exhibits the superconductivity of $NbSe_2$ and the quantum spin Hall effect of monolayer $WTe_2$[11]. Intriguingly, the properties of the different components can be intertwined through interlayer interactions, giving rise to electronic structures and functionalities that surpass those of individual components. Notably, in the 1T-$TaS_2$/1H-$TaS_2$ heterostructure, itinerant electrons from the metallic 1H layer interact with localized moments induced by the charge density waves in the 1T layer, resulting in the emergence of artificial heavy fermions[12].

Traditional approaches to fabricating TMD heterostructures and superlattices typically rely on epitaxial growth or mechanical stacking of 2D materials[12-19]. However, these layer-by-layer construction

[1]International Center for Quantum Design of Functional Materials (ICQD), Hefei National Research Center for Physical Sciences at the Microscale, University of Science and Technology of China, Hefei, Anhui 23026, China. [2]CAS Key Laboratory of Strongly-Coupled Quantum Matter Physics, and Department of Physics, University of Science and Technology of China, Hefei, Anhui 230026, China. [3]Hefei National Laboratory, University of Science and Technology of China, Hefei 230088, China. [4]National Synchrotron Radiation Laboratory, University of Science and Technology of China, Hefei 230029, China. [5]Department of Chemistry, University of Science and Technology of China, Hefei 230029, China. [6]These authors contributed equally: Chaojie Luo, Guohua Cao, Beilin Wang. ✉e-mail: cuipg@ustc.edu.cn; huiz@ustc.edu.cn; cgzeng@ustc.edu.cn

techniques are generally complex and labor-intensive, which constrains the efficiency and quality of fabricating multilayered heterostructures and superlattices. A promising solution to these challenges lies in the exploitation of self-assembly, leveraging the natural propensity of materials to spontaneously organize into ordered structures, thus circumventing the necessity to manually assemble materials block by block. It is noted that prior studies have demonstrated the spontaneous formation of TMD superlattices[5–8,20–31]. For instance, in $TaS_2$ single crystals prepared via the chemical vapor transport (CVT) method, 1T/1H superlattices consisting of alternating stacks of 1T and 1H layers were observed[20,21]. Furthermore, controlling the stoichiometric ratio has been shown to influence the formation energies, thereby obtaining various phases[32,33]. Nonetheless, these superlattices were not deliberately designed but obtained by coincidence, and the formation mechanism behind them remains largely unclear. Therefore, a general approach for the superlattice self-assembly is highly desirable. In this study, we introduce a universal strategy for the self-assembled synthesis of 2D superlattices by balancing formation energies, exemplified by $NbSe_{2-x}Te_x$ 1T/1H superlattice. We demonstrate that adjusting the stoichiometric ratio can balance the formation energies of the T and H phases, enabling the formation of 1T/1H superlattices. Furthermore, we validate the universality of this strategy in two other systems: $Nb_{1-x}V_xSe_2$ and $Nb_{1-x}Ti_xSe_2$.

## Results

### Structure evolution in $NbSe_{2-x}Te_x$ single crystals

$NbSe_2$ and $NbTe_2$ are typical TMD materials, with the former stabilized in the 2H type and the latter stabilized in the 1T'' type in bulk form[34]. Therefore, adjusting the stoichiometric ratio of Te to Se can change the thermal stability of the T phase and H phase in ternary $NbSe_{2-x}Te_x$[35]. Ternary $NbSe_{2-x}Te_x$ with varying Te concentrations ($x = 0.00–2.00$) has been synthesized through CVT in a one-step process. The stoichiometric ratio of the samples was determined by energy dispersive spectroscopy (EDS), as depicted in Supplementary Fig. 1. Figure 1a displays three distinct single crystal morphologies for different Te concentrations of $x = 0.28$, 0.89, and 1.48, corresponding to the 2H, 1T/1H superlattice, and distorted 1T types, respectively, as will be discussed later. Notably, $NbSe_{2-x}Te_x$ single crystals with $0.64 \le x \le 0.89$ exhibit larger thickness along the c-axis compared to other Te concentrations.

Powder X-ray diffractometer (XRD) was employed to characterize the $NbSe_{2-x}Te_x$ single crystals across the Te concentration spectrum, as seen in Fig. 1c. When the Te concentration increases, the typical diffraction peaks shift towards lower angles. Moreover, the $c/n$ (c-axis lattice parameter divided by the number of layers in the stacking repeat), which represents a reduced lattice parameter and signifies layer distance, largely increases with Te concentration (see Fig. 1d). Nevertheless, three regions showing different trends in the evolution of $c/n$ with varying Te concentration are observed, with the $c/n$ remaining almost constant with Te concentration within the range of $0.64 \le x \le 0.89$. The observed overall increase can be primarily attributed to the larger ionic radius of Te, which expands the layer distance in $NbSe_{2-x}Te_x$[32].

To precisely characterize the structures of $NbSe_{2-x}Te_x$ single crystals with different Te concentrations, we performed atomic-scale structural analysis using aberration-corrected scanning transmission electron microscopy (STEM) and scanning tunneling microscopy (STM). As elaborated in Supplementary Note 1, for $x < 0.5$, 1H layers are stacked in an AA' sequence (see Supplementary Fig. 2), designated as the 2H type. When $x > 1.1$, 1T layers adopt an AA stacking sequence (see Supplementary Fig. 2), and exhibit an in-plane 1×3 reconstruction (see Supplementary Fig. 3), manifesting as the distorted 1T type. Figure 1b schematically shows the crystal structures for different types of $NbSe_{2-x}Te_x$ single crystals. Additionally, it is noted that mixed phases can exist between these different structural types, as detailed in Supplementary Note 7.

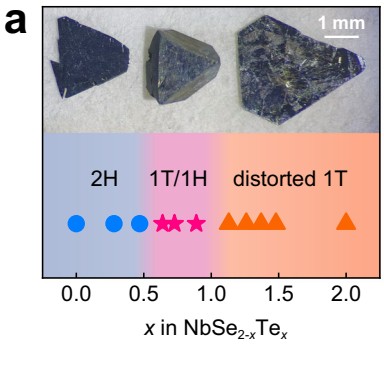

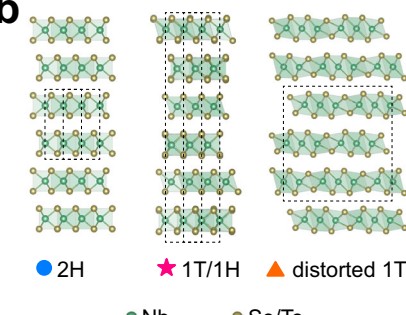

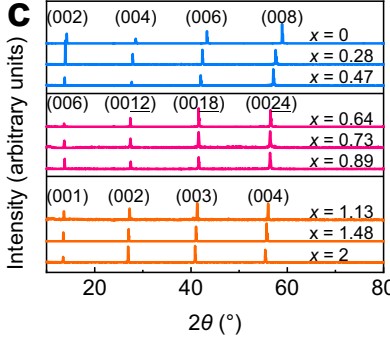

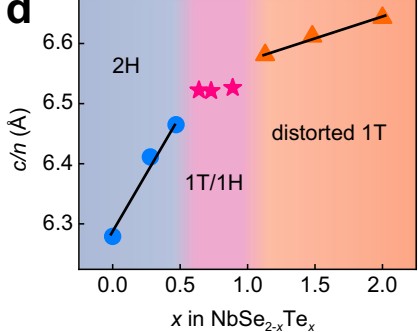

**Fig. 1 | Structure evolution in $NbSe_{2-x}Te_x$ single crystals. a** Structures and morphologies of $NbSe_{2-x}Te_x$ single crystals with varying Te concentrations. **b** Crystal structures of different types of $NbSe_{2-x}Te_x$ single crystals: 2H, 1T/1H superlattice (6R), and distorted 1T (featuring a 1×3 reconstruction) types. The dashed lines indicates the unit cell. **c** X-ray diffraction (XRD) characterization of $NbSe_{2-x}Te_x$ single crystals with different Te concentrations. **d** Reduced lattice parameter ($c/n$) as a function of Te concentration ($x$). The solid black lines show the trends in the evolution of $c/n$ with varying Te concentration.

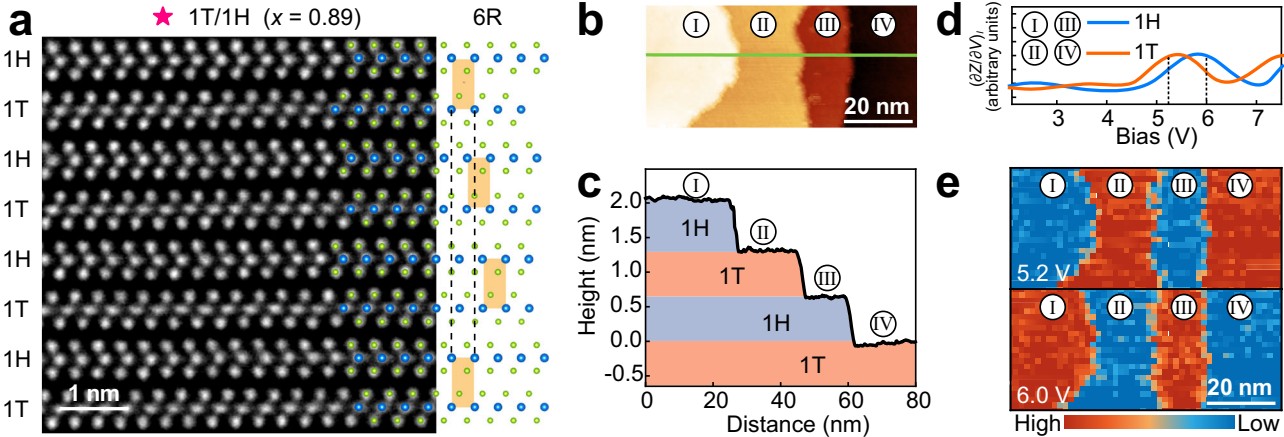

**Fig. 2 | Structure and work functions of the 1T/1H superlattice. a** Cross-sectional high-resolution scanning transmission electron microscopy (STEM) image of the 1T/1H superlattice ($x = 0.89$) along the [110] direction, showing the alternating arrangement of the 1H and 1T layers, superimposed with the 6R atomic structure model. In the model, Nb atoms are represented by blue spheres, while Se or Te atoms are denoted by green spheres. The orange rectangle and the black dash lines show that each 1T-1H heterolayer is slightly displaced along the $c$-axis. **b** Constant-current scanning tunneling microscopy (STM) topography image of terraces measured at 77 K. Scanning parameters: bias voltage $V_{bias} = 1$ V, setpoint current $I_{set} = 100$ pA. **c** Line profile along a cut indicated by the green line in (**b**). **d** The spectra of the derivative of tip height with respect to bias voltage under constant current, $(\partial Z / \partial V)_I$, were measured on the 1T and 1H terminations at 77 K, indicating the differences in the work functions of 1H and 1T layers. **e** $(\partial Z / \partial V)_I$ mapping taken at 5.2 V (upper panel) and 6.0 V (lower panel) for the same region shown in (**b**).

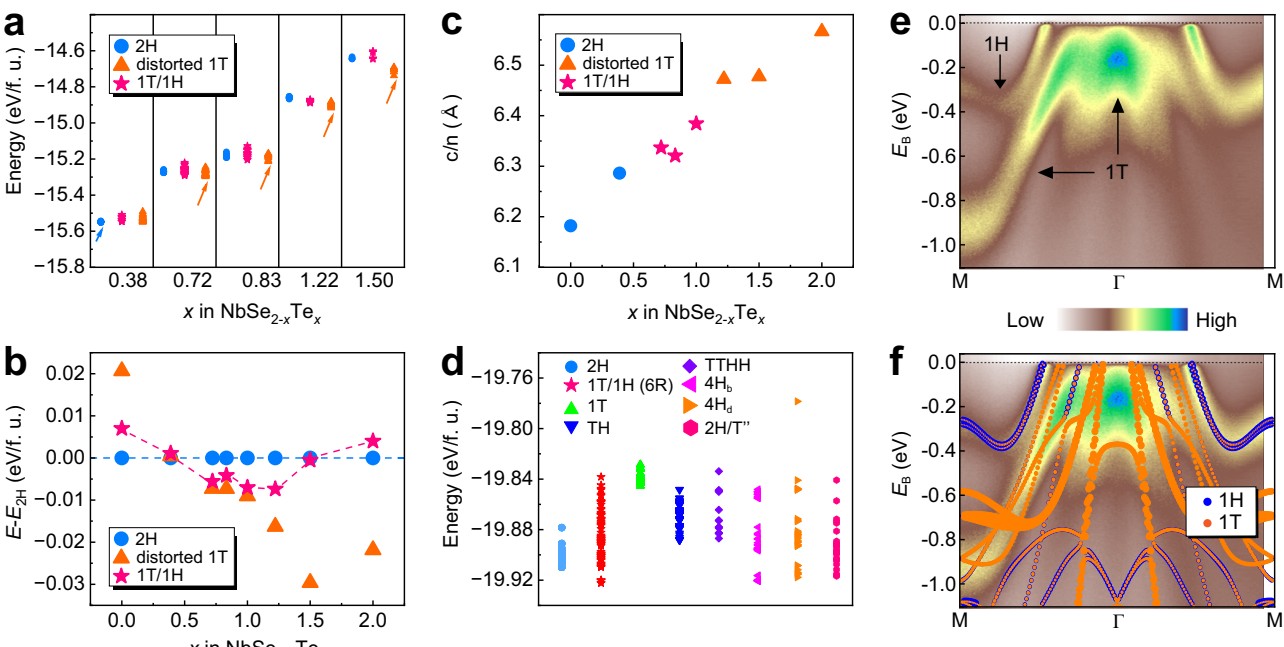

**Fig. 3 | Formation energies, lattice parameters, and electronic structures of NbSe$_{2-x}$Te$_x$. a** Formation energies of 2H, 1T/1H superlattice, and distorted 1T types of NbSe$_{2-x}$Te$_x$ at different Te concentrations ($x$), obtained from density functional theory (DFT) calculations. The most stable doping configuration at each Te concentration is indicated by an arrow. **b** Relative formation energy of the most stable 2H, 1T/1H superlattice, and distorted 1T types as a function of $x$, taking the energy of the 2H type as a reference at each Te concentration. **c** Calculated reduced lattice parameter ($c/n$) as a function of $x$. **d** Formation energies of different stacking configurations at $x = 0.83$. **e** Angle-resolved photoemission spectroscopy (ARPES) spectra along the $\Gamma-M$ direction of the 1T/1H superlattice with $x = 0.73$, taken using 23 eV photons at $T = 10$ K. The black arrows mark the band features originating from the 1H and 1T layers, respectively. **f** Same ARPES spectra as (**e**), but with the calculated layer-resolved band structure of the 1T/1H superlattice with $x = 0.72$ superimposed.

## Structure and work functions of the 1T/1H superlattice

Then, we focus on the NbSe$_{2-x}$Te$_x$ single crystals with $0.64 \leq x \leq 0.89$, which have been shown to exhibit distinct crystal morphology and evolution trend of $c/n$. Figure 2a shows the STEM image of NbSe$_{2-x}$Te$_x$ ($x = 0.89$) viewed along the [110] direction, showcasing the single crystal as a 1T/1H superlattice. This structure is inherently composed of alternating 1T and 1H layers, with each 1T-1H heterolayer shifting by one-third of the lattice constant along the [1$\bar{1}$0] direction, also known as the 6R structure[34]. We measured the Nb-Nb atomic distances in the 1H and 1T layers and found that the atomic distances in both layers are identical, approximately 0.34 nm. The large scale atomic resolution STEM image further reveals that this precise alternation of 1T and 1H

layers extends over a considerably large spatial space, confirming the highly ordered structure of the self-assembled 1T/1H superlattice (see Supplementary Fig. 4). More STEM analyses across single crystals with different Te concentrations reveal that $NbSe_{2-x}Te_x$ single crystals form 1T/1H superlattices for $0.64 \leq x \leq 0.89$ (see Supplementary Fig. 5). Additionally, chemical mappings via energy dispersive X-ray spectroscopy confirm the uniform doping of Te atoms within both 1T and 1H layers in the 1T/1H superlattices (see Supplementary Fig. 6).

We further utilized STM to investigate the in-plane structure and electronic properties of the 1T and 1H layers within the 1T/1H superlattice. The topography image in Fig. 2b reveals the terraces of the 1T/1H superlattice ($x = 0.89$). Figure 2c shows the corresponding height profile, indicating a single-layer step height of approximately 0.66 nm, consistent with the measurements obtained from STEM (see Fig. 2a). From the atomic-resolution STM images in Supplementary Fig. 7, the 1T layers are characterized by the star-of-David reconstruction (see Supplementary Note 2)[19,36], allowing us to identify the terraces: terraces I and III as 1H layers, and terraces II and IV as 1T layers.

Figure 2d displays the $(\partial Z/\partial V)_I$ spectra measured at the 1H and 1T layers, respectively. The peaks in these spectra reflect the field emission resonances (FERs), and the bias voltage corresponding to the first FER peak is considered a good approximation of the work function[37]. The measured work functions of the 1H (5.83 eV) and 1T (5.4 eV) layers align well with those of bulk 2H-$NbSe_2$ (5.9 eV)[38] and 1T''-$NbTe_2$ (5.32 eV)[39], respectively. The observed difference in work function between the 1H and 1T layers suggests a charge transfer from the 1T to 1H layers (see Supplementary Note 3). Figure 2e shows $(\partial Z/\partial V)_I$ mapping at two different voltages (5.2 V and 6.0 V) for the same region depicted in Fig. 2b. At 6.0 V, the mapping indicates that the intensity of the 1H layers (terraces I and III) is greater than that of the 1T layers (terraces II and IV). Conversely, at 5.2 V, the intensity of the 1T layers is higher than that of the 1H layers. Such behaviors provide compelling evidence of the alternating stacking pattern of the 1T and 1H layers in the 1T/1H superlattice.

## Formation mechanism of the 1T/1H superlattices

To gain insights into the growth behavior of the 1T/1H superlattice in $NbSe_{2-x}Te_x$, we systematically investigated the formation energies of three different bulk types of $NbSe_{2-x}Te_x$ with varying Te concentrations using density functional theory (DFT) calculations. The three types are formed by stacking 1H or/and 1T layers, including the 2H, 1T/1H superlattice, and distorted 1T types (see Fig. 1b). The formation energy ($E_f$) is defined as $E_f = E(NbSe_{2-x}Te_x) - \mu(Nb) - (2-x)\mu(Se) - x\mu(Te)$, where $E(NbSe_{2-x}Te_x)$ represents the total energy of $NbSe_{2-x}Te_x$ per formula unit, while $\mu(Nb)$, $\mu(Se)$, and $\mu(Te)$ denote the chemical potentials of Nb, Se, and Te atoms in gas phase, respectively. For each type, the Te concentration was chosen to be $x = 0.0, 0.39, 0.5, 0.72, 0.83, 1.0, 1.22, 1.5$, and $2.0$. To obtain the most stable doping configuration for each fractional stoichiometric $NbSe_{2-x}Te_x$ ($x = 0.39, 0.72, 0.83, 1.22$, or $1.5$), dozens of random configurations were also considered for each type, as shown in Fig. 3a.

Figure 3b summarizes the relative energies of the three types with their respective most stable doping configuration, taking the energy of the 2H type as a reference at each Te concentration. For $x < 0.4$, the 2H type is shown to be the most stable, while for $x > 1.0$, the distorted 1T type becomes the most stable, showing good agreement with the experimental findings (see Fig. 1). For $0.72 \leq x \leq 1.0$, the formation energies of the 1T/1H and distorted 1T types are quite close, both noticeably lower than those of the 2H type, indicating a significant increase in the stability of the 1T/1H superlattice relative to the 2H type. Although the star-of-David reconstruction of the 1T layers has been experimentally observed in the 1T/1H superlattices, it was not considered in our DFT calculations due to computational demands. To partially assess the impact of these reconstructions, we calculated the energy gains for a freestanding monolayer 1T phase and a simplified

1T/1H superlattice model (see Supplementary Fig. 15 for the exact structure), both incorporating star-of-David reconstructions in their 1T layers, as detailed in Supplementary Note 6. The energy gains for the monolayer 1T phase (6.3 meV per formula unit) and the simplified 1T/1H superlattice model (17.7 meV per formula unit) suggest that the presence of star-of-David reconstructions can substantially reduce the formation energy of the 1T/1H superlattice, making it significantly lower than that of the distorted 1T type within the range of $0.64 \leq x \leq 0.89$. Moreover, we examined the average layer distances ($c/n$) for the three types, as shown in Fig. 3c, whose trend also aligns well with the experimental results (Fig. 1d).

Based on the above analyses, it is evident that only 1H or 1T layers are present during the growth process as the doping concentration $x$ approaches 0 or 2.0, due to significant differences in formation energies between the T and H phases. By adjusting the stoichiometric ratio of Te to Se, the formation energy disparity between the T and H phases is reduced, enabling the stable coexistence of 1T and 1H layers during growth. To explore the underlying formation mechanism of the 1T/1H superlattice in $NbSe_{2-x}Te_x$, we constructed eight bulk configurations with different stacking orders of 1H or/and 1T layers at $x = 0.83$, as shown in Supplementary Fig. 8. Here, the 6R configuration corresponds to the experimentally observed 1T/1H superlattice. Again, with dozens of random doping configurations considered, the formation energies for all configurations are illustrated in Fig. 3d. Notably, the formation energies of $4H_b$, $4H_d$, and 6R configurations are close and significantly lower than those of other stacking configurations. Among them, the 6R configuration is the most stable, consistent with the experimental observation of the 1T/1H superlattice. While the $4H_b$, $4H_d$, and 6R configurations all feature alternating 1T and 1H layers, the distinct stacking orders suggest that interlayer interactions also play a role in the formation of the 1T/1H superlattice. More details can be found in Supplementary Note 9. Additionally, as presented in Supplementary Note 3, there exists a charge transfer of 0.5 $e$ per unit cell between neighboring 1T and 1H layers (see Supplementary Fig. 9). Similar to the misfit layer compounds[40], such a charge transfer may lead to electrostatic binding between the 1T and 1H layers, eventually resulting in their self-assembly into the 1T/1H superlattice rather than other superlattices.

Next, the electronic structures of the 1T/1H superlattices were investigated using angle-resolved photoemission spectroscopy (ARPES), with $NbSe_{2-x}Te_x$ ($x = 0.73$) as a representative case. The electronic dispersion of the 1T/1H superlattice along the Γ-M direction is presented in Fig. 3e, exhibiting distinctive features of both the 1T and 1H layers. Specifically, a pair of electron-like spin-degenerate bands near the M point are observed, resembling the band structure of monolayer 1H-$NbSe_2$ without interlayer interaction[41,42]. The dispersive hole-like bands centered around the Γ point are similar to the band structure of monolayer 1T-$NbSe_2$, albeit shifted upwards towards the Fermi level[43]. Additionally, based on the first-principles calculations with virtual crystal approximation[44,45] for $NbSe_{2-x}Te_x$ with $x = 0.72$, we further confirm that these two distinct band features around the M and Γ points originate from the 1H and 1T layers, respectively, as shown in Fig. 3f. The ARPES intensity map at the Fermi level is shown in Supplementary Fig. 10a. The Fermi surface displays a hexagonal contour around the Γ point and an electron pocket at the K point, both attributed to the 1H layer[46], while the pronounced density of state at the Γ point is contributed by the 1T layer. Therefore, the band structure of the 1T/1H superlattice possesses a mixed characteristic of the 1H and 1T layers.

## Validating the strategy for superlattice self-assembly

Finally, to further validate the universality of our strategy, we explored to achieve 1T/1H superlattices through the substitution of transition metal atoms. Given that we have obtained 1T/1H superlattices by adjusting the stoichiometric ratio of chalcogen atoms in $NbSe_{2-x}Te_x$,

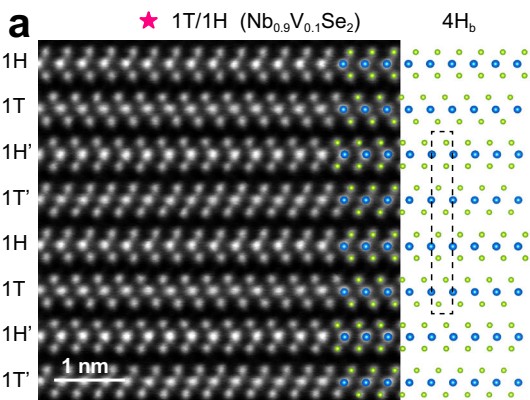

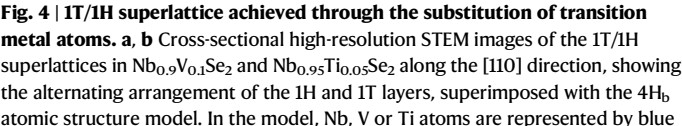

**Fig. 4 | 1T/1H superlattice achieved through the substitution of transition metal atoms. a, b** Cross-sectional high-resolution STEM images of the 1T/1H superlattices in $Nb_{0.9}V_{0.1}Se_2$ and $Nb_{0.95}Ti_{0.05}Se_2$ along the [110] direction, showing the alternating arrangement of the 1H and 1T layers, superimposed with the $4H_b$ atomic structure model. In the model, Nb, V or Ti atoms are represented by blue spheres, while Se atoms are denoted by green spheres. The black dashed lines indicate that all transition metal atoms in each 1T-1H heterolayer are aligned along the same axis parallel to the c-axis, while the adjacent 1T-1H heterolayers rotate by 60° around a central transition metal atom.

we now investigate whether substituting metal atoms could yield similar results. Considering that bulk $NbSe_2$ stabilizes in the 2H type, while both $VSe_2$ and $TiSe_2$ stabilize in the 1T type, we hypothesize that substituting Nb atoms in $NbSe_2$ with V or Ti atoms would reduce the formation energy disparity between the T and H phases, thereby facilitating the formation of a 1T/1H superlattice.

To test our hypothesis, we synthesized $Nb_{1-x}V_xSe_2$ and $Nb_{1-x}Ti_xSe_2$ single crystals using a one-step CVT process. The XRD and EDS results for $Nb_{0.9}V_{0.1}Se_2$ and $Nb_{0.95}Ti_{0.05}Se_2$ are displayed in Supplementary Fig. 11, and the STEM images viewed along the [110] direction are shown in Fig. 4. Notably, both single crystals exhibit the 1T/1H superlattice structure. All the transition metal atoms in each 1T-1H heterolayer are aligned along the same axis parallel to the c-axis, while adjacent 1T-1H heterolayers rotate by 60° around a central transition metal atom, forming the $4H_b$ structure. These findings not only validate the universality of our strategy but also suggest a promising route for tuning the properties of 1T/1H superlattice through stacking engineering.

In summary, we introduce a universal strategy for the self-assembled synthesis of 1T/1H superlattices in TMDs by balancing the formation energies of the 1T and 1H phases, showcased by $NbSe_{2-x}Te_x$ single crystals. By precisely tuning the stoichiometric ratio of Te to Se in $NbSe_{2-x}Te_x$, the difference in formation energies of the T and H phases is significantly reduced, enabling the spontaneous stacking of 1T and 1H layers to form 1T/1H superlattices. These 1T/1H superlattices not only exhibit an alternating stacking pattern of the 1T and 1H layers but also feature an electronic structure that merges the characteristics of both layers. The universality of this strategy is further validated by achieving 1T/1H superlattices in $NbSe_2$ through substituting Nb with V or Ti atoms. Our approach simplifies the fabrication of superlattices and could be extended to other TMDs, offering a new pathway for exploring their properties and applications in electronic devices.

## Methods

### Synthesis of $NbSe_{2-x}Te_x$ single crystals

Single crystals of $NbSe_{2-x}Te_x$ were prepared by the method of CVT with iodide as the transport agent. Stoichiometric amounts of high purity Nb, Te, and Se with a total weight of 1.5 g and 220 mg iodide were sealed in an evacuated 20 cm long quartz tube under vacuum at $10^{-4}$ Torr. The reaction zone was then programmed at a higher temperature of 850 °C with the growth zone at a lower temperature of 800 °C for 7 days. Then the quartz tubes were removed from the furnace and quenched in the ice water mixture and the single crystal of $NbSe_{2-x}Te_x$ can be collected in the growth zone.

### Sample characterization

XRD patterns were collected using a Rigaku SmartLab SE X-ray diffractometer with Cu Kα radiation ($\lambda = 0.15418$ nm) at room temperature. Scanning electron microscopy (SEM) and X-ray energy-dispersive spectroscopy (EDS) were performed using a HITACHI S5000 with a Bruker XFlash 6 | 60 energy dispersive analysis system.

### STEM sample preparations and STEM measurements

STEM samples were prepared by a dual beam focused ion beam (FEI Helios Nanolab G3) using standard lift-out procedures. The STEM images and EDS mapping images were acquired using a CEOS probe corrected FEI Themis Z at an electron accelerating voltage of 300 kV with a probe convergence angle of 17.8 mrad, spatial resolution of 0.08 nm, and probe current of ~ 60 pA for STEM imaging and ~ 130 pA for EDS mapping. The collection angle range for the high angle annular dark field detector is 40–200 mrad. STEM images were filtered by the standard high-pass filtering method to reduce noise.

### STM measurements

STM experiments were carried out in a commercial Unisoku STM USM-1300 operated at 77 K in ultrahigh vacuum (better than $3 \times 10^{-10}$ mbar). A tungsten tip was used and calibrated on Au(111). Samples were cleaved in ultra-high vacuum at room temperature. The scanning parameters of the STM topographic images are listed in the figure captions. The $(\partial Z/\partial V)_I$ spectrum was acquired when the tip-to-sample distance Z changes corresponding to the scanning of bias V to keep the constant current with the feedback loop on.

### ARPES measurements

ARPES experiments were performed at BL13U at the National Synchrotron Radiation Laboratory (NSRL) equipped with a Scienta DA30-L analyzer. The crystals were cleaved in situ below 20 K and measured in an ultrahigh vacuum with a base pressure better than $6.0 \times 10^{-11}$ mbar. The overall energy resolution was better than 30 meV, and the angular resolution was 0.3°. The Fermi level was calibrated utilizing the Fermi edge of polycrystalline gold.

### DFT calculations

All first-principles calculations were performed within DFT implemented in the Vienna ab initio simulation package (VASP)[47] using the generalized gradient approximation of Perdew, Burke and Ernzerhof[48] as the exchange-correlation functional. For VASP calculations, a 6×6×3 Monkhorst-Pack k-mesh was used to sample the first Brillouin zone[49].

The core electrons were treated fully relativistically by the projector augmented wave method[50], while the valence electrons were processed in the scalar relativistic approximation, with a plane-wave cutoff of 500 eV. All the atoms were allowed to fully relax during structural optimization until all the forces on each atom were less than 0.01 eV/Å.

## Data availability

The source data of the main figures in this study have been deposited in the Figshare database under accession code https://doi.org/10.6084/m9.figshare.27850653.v2. All raw data generated during the current study are available from the corresponding authors upon request.

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

## Acknowledgements

C.L., G.C., B.W., L.J., H.Y.Z., T.L., X.T., Z.L., Y.L., Z.S., P.C., H.Z., Z.Z. and C.Z. were supported by the National Key Research and Development Program of China (grant no. 2023YFA1406300), the Innovation Program for Quantum Science and Technology (grant no. 2021ZD0302800), the National Natural Science Foundation of China (grant nos. 92165201, 12074357, 12004368, 12374458, and 11974323), Anhui Provincial Key Research and Development Project (grant no. 2023z04020008), the CAS Project for Young Scientists in Basic Research (grant no. YSBR-046), the Fundamental Research Funds for the Central Universities (grant nos. WK9990000118 and WK2310000104), and the Strategic Priority Research Program of Chinese Academy of Sciences (grant no. XDB0510200).

## Author contributions

C.Z. and H.Z. designed and supervised the work. C.L. and B.W. performed the experiments with assistance from L.J., H.Y.Z., T.L., X.T., Y.L., Z.L. and Z.S.; G.C., P.C. and Z.Z. conceived the theoretical model. C.L., H.Z., G.C., P.C. and C.Z. analyzed the data and wrote the manuscript. All authors contributed to the scientific discussion and manuscript revisions.

## Competing interests

The authors declare no competing interests.
