## [Transparent Peer Review file · Nature Communications]

Self-assembly of 1T/1H superlattices in transition metal dichalcogenides

Corresponding Author: Professor Changgan Zeng

Version 0:

Reviewer comments:

Reviewer #1

(Remarks to the Author)
See attachment.

Reviewer #2

(Remarks to the Author)
I co-reviewed this manuscript with one of the reviewers who provided the listed reports. This is part of the Nature Communications initiative to facilitate training in peer review and to provide appropriate recognition for Early Career Researchers who co-review manuscripts.

Reviewer #3

(Remarks to the Author)
The manuscript by Chaojie Luo et al. presents an approach for the automated synthesis of TMD superlattice single crystals through self-assembly. By adjusting the Te to Se stoichiometric ratio in $\text{NbSe}_{2-x}\text{Te}_x$, the authors balance the formation energy disparity between the T and H phases, enabling the self-assembly of 1T and 1H layers into a 1T/1H superlattice. They also successfully achieved 1T/1H superlattices in NbSe_2 by substituting Nb with V or Ti atoms. The authors propose that this work introduces a universal strategy for the self-assembly synthesis of 1T/1H superlattices in TMDs, with potential applicability across a broad range of TMD materials.

While the findings could be of interest to researchers in the field of TMDs, the novelty of this work, in its current form, may not meet the high standards expected by Nature Communications. Firstly, the authors seem to overstate the novelty of their work. Stoichiometric control has already been demonstrated to fabricate heterostructures comprising different phases due to their distinct formation energies (see Proceedings of the National Academy of Sciences 112.11 (2015): E1174-E1180; Nature Communications 15.1 (2024): 2541). Additionally, the 1T/1H (6R) superlattice observed in this study has been reported in previous research as well (arXiv:2402.07609v1, published on 12 Feb 2024). Therefore, this work may not represent a significant fundamental or methodological advance over existing studies, despite the high quality of the STEM and STM results presented.

Moreover, some of the conclusions drawn in the manuscript require more comprehensive experimental validation. For instance, the authors claim that $\text{NbSe}_{2-x}\text{Te}_x$ single crystals exhibit a 2H type when $x < 0.5$ and transition to a distorted 1T type (with 1×3 reconstruction) when $x > 1.1$. However, only one data point was studied for each case ($x=0.28$ and $x=1.48$). This conclusion also conflicts with previous work showing that NbSeTe ($x=1$) is in the 1T phase (Journal of Physics: Condensed Matter 32.2 (2019): 025702). Additional experimental studies are necessary to substantiate this conclusion. Lastly, the manuscript generally lacks detailed information about the experiments, such as EELS, which is crucial for validating the presented findings.

Reviewer #4

(Remarks to the Author)

Manuscript title: Self-assembly of 1T/1H superlattice in transition metal dichalcogenides

Manuscript ID: NCOMMS-24-45261

Heterostructures and superlattices composed of layered transition metal dichalcogenides (TMDs) benefit from their superior emergent properties over individual components and provide significant promise for the development of multifunctional electronic devices. However, traditional synthesis methods for these structures depend on layer-by-layer artificial construction and are limited by their complexity and time-consuming. In this work, the authors develop a universal strategy for the automated synthesis of TMD superlattice single crystals through self-assembly, exemplified by the NbSe₂-xTex 1T/1H superlattice, which retains the electronic characteristics of both 1T and 1H layers. This self-assembly for superlattice crystal synthesis approach could extend to other TMDs (such as Nb_{1-x}(V/Ti)_xSe₂) or layered materials, opening new avenues for efficient fabrication and broad applications of superlattices. Therefore, this simple method is helpful for the efficient development of new heterostructure superlattices and novel physical properties. It can be accepted after addressing the following comments:

(1) Since the typical TMD NbSe₂ and NbTe₂ usually stabilize in the 2H type and 1T'' type, respectively, why does this NbSe₂-xTex superlattice trend to stabilize in the 1T/1H superlattice, not 2H/1T'' or other superlattice? What is the principle factor for the stabilization of the 1T/1H superlattice?

(2) The authors claimed that some previous reports that adjusting the stoichiometric ratio of Te to Se can change the thermal stability of the T phase and H phase in ternary NbSe₂-xTex, such as the previous adjusting ratio of Te to Se in the NbSe₂-xTex to 1 could reach 1T-NbSeTe (e.g., J. Phys.: Condens. Matter 2020, 32 025702). From Figure 1a, it can be seen that 1T/1H NbSe₂-xTex 1T/1H superlattice was formed. Why does this new method cause the formation of NbSe₂-xTex 1T/1H superlattice when x = 1, not pure 1T type? It is better to give more explanation.

(3) Figure 1d shows three types of crystal structures in these ternary NbSe₂-xTex compositions, which are distinguished by color regions. Is it curious there are no mixing phases between these regions?

(4) Are there unique physical properties, such as superconductivity, for this 1T/1H NbSe₂-xTex 1T/1H superlattice?

Version 1:

Reviewer comments:

Reviewer #1

(Remarks to the Author)

I am satisfied with the revisions made in the latest version.

Reviewer #2

(Remarks to the Author)

Reviewer #3

(Remarks to the Author)

The authors have adequately addressed the concerns initially raised. Given that the self-assembly of the 1T/1H superlattice is primarily influenced by the equilibrium between the formation energies of the various phases, it would be beneficial to include a more comprehensive introduction, properly citing the two papers that were discussed in the initial review on the stacking of 1H, 2H, and 3R types. Once these revisions have been made, I am satisfied with publishing this work.

Reviewer #4

(Remarks to the Author)

The authors have addressed all my questions, it can be accepted now.

Responses to the reviewers

(Manuscript NCOMMS-24-45261)

Detailed responses to reviewer #1:

General Comment: *The authors present a universal approach for the automated synthesis of TMD superlattice single crystals via self-assembly, demonstrated using the $\text{NbSe}_{2-x}\text{Te}_x$ 1T/1H superlattice. The central concept of this method lies in balancing the formation energies of the T (octahedral) and H (trigonal prismatic) phases. By fine-tuning the Te to Se stoichiometric ratio in $\text{NbSe}_{2-x}\text{Te}_x$, the authors successfully minimize the formation energy difference between the T and H phases, thereby facilitating the self-assembly of 1T and 1H layers into a coherent 1T/1H superlattice. The authors initially focused their study on Te-doped NbSe_2 and subsequently extended their investigation to include V-doped and Ti-doped NbSe_2 systems.*

Response: We appreciate the reviewer's valuable feedback and suggestions. As shown in the response below, we have addressed the comments raised by the reviewer, and have revised the manuscript accordingly.

Comment 1: *One technique concern lies with the X-ray diffractometer technique employed for structural determination. The authors have described the instrument as a single crystal X-ray diffractometer. However, in the experimental section, they state, 'XRD patterns were collected using a Rigaku SmartLab SE X-ray diffractometer with Cu K α radiation ($\lambda = 0.15418$ nm) at room temperature.' It is important to note that the Rigaku SmartLab SE is a powder X-ray diffractometer. Therefore, I recommend that the authors revise their description to accurately reflect this by stating that a 'powder X-ray diffractometer was applied to single crystal samples.'*

Response: We thank the reviewer for pointing out our inaccurate expression, and we have updated the description on Line 72 of the revised manuscript following the reviewer's suggestion.

Comment 2: *Let's closely examine the structural 'models' presented. I refer to them as 'models' because it is unclear how the authors determined the precise atomic coordinates using only powder X-ray diffraction and STEM techniques. The figure below corresponds to Fig. 1b in the main text. The structural differences between the 2H and 3R phases can be simplified as the packing of $\text{Nb}@\text{(Se/Te)}_6$ trigonal prisms or octahedral. In the 2H phase, the $\text{Nb}@\text{(Se/Te)}_6$ trigonal prisms are packed in alternating orientations, whereas in the 3R phase, the $\text{Nb}@\text{(Se/Te)}_6$ trigonal prism layers are interspersed with layers of octahedral $\text{Nb}@\text{(Se/Te)}_6$.*

My first question concerns the Se/Te atomic mixture. How did the authors determine the atomic occupancy for each site? For instance, in the 3R phase, the authors labeled the octahedral layers as containing only Nb@Se₆. What method was used to determine this—STEM, perhaps? If it is true that Se fully occupies the octahedral layers, an inconsistency arises: how can the lattice parameters remain constant as the Te doping amount increases in the 1T/1H phases?

Response: We thank the reviewer for this comment. Based on the statement, "whereas in the 3R phase, the Nb@(Se/Te)₆ trigonal prism layers are interspersed with layers of octahedral Nb@(Se/Te)₆," it seems that the reviewer might have intended to refer to the 6R phase instead of the 3R phase. This interpretation arises because, in the 3R phase, the Nb@(Se/Te)₆ trigonal prisms are arranged in a sequentially offset manner, while in the 6R phase, trigonal prism layers alternate with octahedral layers of Nb@(Se/Te)₆ (Phys. Rev. B 69, 13 (2004)). Therefore, our response will focus on the 6R phase observed in NbSe_{2-x}Te_x.

We utilized STEM to determine the structure and employed STEM-EDS mapping to identify the occupancy of various elements, as illustrated in the original Supplementary Fig. 6. Figures S6d and S6e depict the chemical mappings of Se and Te atoms, respectively. The spatial distribution of Se and Te overlaps, with their intensities in the trigonal prismatic and octahedral layers being essentially identical. This suggests that Se and Te are randomly distributed within both the trigonal prismatic and octahedral layers. As the reviewer pointed out, our initial model inaccurately represented the distribution of Se and Te across different layers. The model misleadingly depicted Se and Te atoms in different colors, suggesting an uneven distribution that was not intended.

We have now updated the original model in Fig. 1b with a revised version (Fig. R1), where Se and Te atoms are represented in the same color to accurately reflect their random distribution.

Fig. R1 | Crystal structures of different types of NbSe_{2-x}Te_x single crystals: 2H, 1T/1H superlattice (6R), and distorted 1T (featuring a 1×3 reconstruction) types.

Comment 3: *Let's now examine the STEM image presented in Fig. 2a. Could the authors specify the metal-to-metal atomic distances, specifically the Nb-Nb distance, in the 1H and 1T phases separately? Additionally, it appears that the electron distribution in the 1T layer shows a dumbbell-like shape, I marked in red in the figure below. Could the authors provide more insight or confirmation regarding this observation?*

Response: We appreciate the reviewer for this comment. We respond to this comment in two aspects:

(1) Following the reviewer's suggestion, we measured the Nb-Nb atomic distances in the 1H and 1T layers and found that the atomic distances in both layers are identical, approximately 0.34 nm. These results have been included in the revised manuscript (Lines 97-98).

(2) We speculate that the distinct dumbbell-like electron distribution observed in the 1T layer results from reconstruction-induced lateral local deviations of some Nb atoms along the [110] direction. As shown in original Supplementary Fig. 7, we identified notable complex reconstructions within the 1T layers and recognized short-range star-of-David reconstructions, whereas no such reconstruction exists in the 1H layer. Figure

R2a confirms the presence of these reconstructions at room temperature in the 1T layer. Since it is challenging to describe such complex reconstructions, our model only considers the star-of-David reconstructions. Figures R2b and R2c depict an atomic model of a $\text{NbSe}_{2-x}\text{Te}_x$ 1T layer with the star-of-David reconstruction, clearly exhibiting lateral local deviations of the Nb atoms. Based on this evidence and analysis, these reconstructions in the 1T layer are responsible for the observed dumbbell-like distribution of the Nb atomic column in the $\text{NbSe}_{2-x}\text{Te}_x$ 1T/1H superlattices.

In the revision, we have included Fig. R2 as the new Supplementary Fig. 12, with relevant discussions as the new Supplementary Note 5.

Fig. R2 | Reconstruction of the 1T layer in 1T/1H superlattice ($\text{NbSe}_{2-x}\text{Te}_x$) at room temperature. **a** Topography of a cleaved surface displaying 1T layer termination in the 1T/1H superlattice at $T = 300$ K. Scanning parameters: $V_{\text{bias}} = 0.01$ V, $I_{\text{set}} = 2.5$ nA. **b** Top view of the 1T layer featuring the star-of-David reconstruction. **c** Lateral view of the 1T layer with star-of-David reconstruction along the $[110]$ direction.

Comment 4: *The energy difference shown in Fig. 3 between the 1T/1H and distorted 1T phases is quite small. In fact, the distorted 1T phase exhibits a lower energy than the 1T/1H phase at around $x(\text{Te}) \sim 0.8$. Given this, how can the authors assert that the formation energy difference between the T and H phases is the primary factor controlling phase stability?*

Response: We thank the reviewer for this insightful comment. From our DFT calculations, it is indeed observed that the distorted 1T type exhibits slightly lower energies by 1.7 and 3.1 meV per formula unit for $x = 0.72$ and $x = 0.83$, respectively, compared to the corresponding 1T/1H superlattice. However, it is important to note that, because the computations were too demanding, the star-of-David reconstructions observed in the 1T layers of the 1T/1H superlattice (see original Supplementary Fig. 7) were not included in our DFT calculations. To partially assess the impact of these reconstructions, we calculated the energy gains for a freestanding monolayer 1T phase and a simplified 1T/1H superlattice model, both incorporating star-of-David reconstructions in their 1T layers, especially for $\text{NbSe}_{2-x}\text{Te}_x$ with $x = 0.77$. In the simplified 1T/1H superlattice model, the 1T layer in each unit cell is shifted by one-third of the lattice constant relative to the 1H layer along the $[1\bar{1}0]$ direction. As

illustrated in Fig. R3, the energy gain for the freestanding monolayer 1T phase is 6.3 meV per formula unit, while this value increases to 17.7 meV per formula unit in the simplified 1T/1H superlattice model. These findings suggest that the presence of star-of-David reconstructions can substantially reduce the formation energy of the 1T/1H superlattice, making it significantly lower than that of the distorted 1T type within the range of $0.64 \leq x \leq 0.89$, thereby facilitating the formation of the 1T/1H superlattice.

In the revision, we have updated the statements in Lines 145-152 and included Fig. R3 as the new Supplementary Fig. 15, with relevant discussions provided in the new Supplementary Note 6.

Fig. R3 | Crystal structures of two different phases in $\text{NbSe}_{2-x}\text{Te}_x$ with $x = 0.77$. Top (left panel) and side (right panel) views of the atomic structures without (a) and with (b) star-of-David reconstruction for a freestanding monolayer 1T phase. c, d Same as (a, b) but for a simplified 1T/1H superlattice model. In the simplified 1T/1H superlattice model, the 1T layer in each unit cell is shifted by one-third of the lattice constant relative to the 1H layer along the $[\bar{1}\bar{1}0]$ direction.

Comment 5: *In Fig. 4, for the V-doped system, the metal atoms at the 1H and 1H' sites appear brighter than those at the 1T and 1T' sites. Did the authors verify which sites have a higher concentration of V atoms? Additionally, this phenomenon is not as clearly observed in the Ti-doped system. Why was the doping concentration of V set at 0.1 and Ti at 0.05? Was there a specific reason for not using the same doping levels for V and Ti?*

Response: We thank the reviewer for his/her valuable comment. We respond to this comment in three aspects:

(1) Following the reviewer's suggestion, we performed STEM-EDS mapping on the $\text{Nb}_{0.86}\text{V}_{0.14}\text{Se}_2$ 1T/1H superlattice, as shown in Fig. R4. The results indicate that V atoms exhibit the same spatial distribution and similar intensity in both the 1T and 1H layers, suggesting that their doping concentrations are essentially identical.

Fig. R4 | Atomic-scale structural and chemical analysis of $4H_b\text{-Nb}_{0.86}\text{V}_{0.14}\text{Se}_2$. **a** Atomic-resolution STEM image acquired simultaneously with chemical mapping via energy dispersive X-ray spectroscopy imaging. **b** Overlay of the Nb (green), Se (red) and V (cyan) signals. **c** Chemical mapping of Nb. **d** Chemical mapping of Se. **e** Chemical mapping of V. **f** Lateral integration of chemical mapping intensities. The integrated intensity curves are shifted horizontally for clarity.

(2) We propose that the appearance of brighter metal atoms at the 1H and 1H' sites, compared to the 1T and 1T' sites, is linked to the reconstruction in the 1T layer. As shown in Fig. R5, both the V-doped and Ti-doped systems exhibit a star-of-David reconstruction in the 1T layer at room temperature. However, this reconstruction is more ordered in the V-doped system than in the Ti-doped system. This may result in the metal atoms at the 1H and 1H' sites to appear brighter than those at the 1T and 1T' sites in the V-doped system.

Fig. R5 | Reconstruction of the 1T layer in 1T/1H superlattice ($\text{Nb}_{1-x}\text{V}_x\text{Se}_2$, $\text{Nb}_{1-x}\text{Ti}_x\text{Se}_2$) at room temperature. **a** Topography of a cleaved surface showing 1T layer termination of $4H_b\text{-Nb}_{0.9}\text{V}_{0.1}\text{Se}_2$ at $T = 300$ K. **b** Topography of a cleaved surface showing 1T layer termination of $4H_b\text{-Nb}_{0.95}\text{Ti}_{0.05}\text{Se}_2$ at $T = 300$ K. Scanning parameters: $V_{\text{bias}} = 0.05$ V, $I_{\text{set}} = 2$ nA.

(3) Based on our experimental results, the approximate concentration ranges for the formation of the 1T/1H superlattices are $0.1 \leq x \leq 0.14$ for $\text{Nb}_{1-x}\text{V}_x\text{Se}_2$ and $0.04 \leq x \leq$

0.05 for $\text{Nb}_{1-x}\text{Ti}_x\text{Se}_2$. The varying doping concentrations required for V and Ti in the 1T/1H superlattices may be attributed to the distinct effects that V and Ti have on the formation energies.

In the revision, we have included Fig. R4 as the new Supplementary Fig. 13 and Fig. R5 as the new Supplementary Fig. 14. Relevant discussions have been added in the new Supplementary Note 5.

Detailed responses to reviewer #2:

General Comment: *I co-reviewed this manuscript with one of the reviewers who provided the listed reports.*

Response: We thank the reviewer for dedicating his/her time and effort to co-review our manuscript.

Detailed responses to reviewer #3:

General Comment: *The manuscript by Chaojie Luo et al. presents an approach for the automated synthesis of TMD superlattice single crystals through self-assembly. By adjusting the Te to Se stoichiometric ratio in $\text{NbSe}_{2-x}\text{Te}_x$, the authors balance the formation energy disparity between the T and H phases, enabling the self-assembly of 1T and 1H layers into a 1T/1H superlattice. They also successfully achieved 1T/1H superlattices in NbSe_2 by substituting Nb with V or Ti atoms. The authors propose that this work introduces a universal strategy for the self-assembly synthesis of 1T/1H superlattices in TMDs, with potential applicability across a broad range of TMD materials.*

Response: We thank the reviewer for his/her careful review of the manuscript. Below we have addressed all the comments raised by the reviewer in detail and have revised the manuscript accordingly.

Comment 1: *While the findings could be of interest to researchers in the field of TMDs, the novelty of this work, in its current form, may not meet the high standards expected by Nature Communications. Firstly, the authors seem to overstate the novelty of their work. Stoichiometric control has already been demonstrated to fabricate heterostructures comprising different phases due to their distinct formation energies (see Proceedings of the National Academy of Sciences 112.11 (2015): E1174-E1180; Nature Communications 15.1 (2024): 2541). Additionally, the 1T/1H (6R) superlattice observed in this study has been reported in previous research as well (arXiv:2402.07609v1, published on 12 Feb 2024). Therefore, this work may not represent a significant fundamental or methodological advance over existing studies, despite the high quality of the STEM and STM results presented.*

Response: We appreciate the reviewer's comment. We respond to this comment in two aspects:

(1) First, we clarify that the first two articles cited by the reviewer investigate stacks of pure H or T phases, rather than heterostructures or 1T/1H superlattices. Specifically, the first article addresses the 2H and 3R types, which are stacks of 1H layers, and the 1T type, comprised of stacks of 1T layers (Proceedings of the National Academy of Sciences 112.11 (2015): E1174-E1180). Similarly, the second article explores the 2H and 3R types stacked with 1H layers (Nature Communications 15.1 (2024): 2541).

(2) The third article cited by the reviewer presents evidence of a 1T/1H superlattice (arXiv:2402.07609v1, published on February 12, 2024), but it is limited to a single phase point in the $\text{NbSe}_{2-x}\text{Te}_x$ phase diagram, namely, $x = 1$, rather than controlled growth of 1T/1H superlattices and the corresponding formation mechanism. In our

study, we propose a universal strategy for synthesizing 1T/1H superlattices, using $\text{NbSe}_{2-x}\text{Te}_x$ as a model system. We demonstrate that adjusting the stoichiometric ratio can balance the formation energies of the T and H phases, enabling the formation of 1T/1H superlattices. Furthermore, we have verified the universality of this strategy in other two systems, namely, $\text{Nb}_{1-x}\text{V}_x\text{Se}_2$ and $\text{Nb}_{1-x}\text{Ti}_x\text{Se}_2$.

In summary, our work provides a general and convenient strategy for synthesizing the challenging 1T/1H superlattices. Compared to previous studies, we advance the field by demonstrating a novel approach to achieve TMD superlattices and thus lay a solid foundation for future investigations into such superlattices. We have revised the introduction (Lines 55-58) to better emphasize the significance of our findings, and we have added the reference (arXiv:2402.07609v1, published on February 12, 2024) mentioned by the Reviewer as ref. 31 in the revised manuscript.

Comment 2: *Moreover, some of the conclusions drawn in the manuscript require more comprehensive experimental validation. For instance, the authors claim that $\text{NbSe}_{2-x}\text{Te}_x$ single crystals exhibit a 2H type when $x < 0.5$ and transition to a distorted 1T type (with 1×3 reconstruction) when $x > 1.1$. However, only one data point was studied for each case ($x=0.28$ and $x=1.48$). This conclusion also conflicts with previous work showing that NbSeTe ($x=1$) is in the 1T phase (Journal of Physics: Condensed Matter 32.2 (2019): 025702). Additional experimental studies are necessary to substantiate this conclusion.*

Response: We thank the reviewer for his/her valuable comment. We respond to this comment in two aspects:

(1) In Fig. 1d of the original manuscript, we demonstrated that $\text{NbSe}_{2-x}\text{Te}_x$ single crystals exhibit a 2H type when $x = 0, 0.28, 0.47$, and transform into a distorted 1T type at $x = 1.13, 1.48$, and 2. In the original Supplementary Information, we did not provide all the necessary STEM results for these concentrations. Following the reviewer's suggestion, we have now included additional STEM data for $x = 0.28, 0.47, 1.13$, and 1.48, as shown in Fig. R6, to reinforce our conclusions regarding the structural phases of $\text{NbSe}_{2-x}\text{Te}_x$ single crystals.

Fig. R6 | Atomic-scale structures of 2H and distorted 1T types. a, b STEM images of 2H structures with $x = 0.28$ and 0.47 . **c, d** STEM images of distorted 1T structures with $x = 1.13$ and 1.48 .

(2) To clarify the structure of $\text{NbSe}_{2-x}\text{Te}_x$ at $x = 1$, we grew new $\text{NbSe}_{2-x}\text{Te}_x$ single crystals and conducted STEM experiments. As shown in Fig. R7, the $\text{NbSe}_{2-x}\text{Te}_x$ single crystal at $x = 1.03$ comprises a mixture of a distorted 1T type (marked by the red regions) and a 1T/1H superlattice (marked by the blue regions). These results suggest that when $x \approx 1$, $\text{NbSe}_{2-x}\text{Te}_x$ is in a transitional phase between the 1T/1H superlattice and distorted 1T type. Therefore, it is reasonable that the previous work (Journal of Physics: Condensed Matter 32.2 (2019): 025702) reported NbSeTe as a 1T type structure at $x = 1$, since different growth conditions may lead to structural variations for the transitional phase.

Fig. R7 | Atomic-scale structure of NbSe_{2-x}Te_x (x = 1.03). STEM images of NbSe_{2-x}Te_x (x = 1.03). The red regions mark the distorted 1T type and the blue regions mark the 1T/1H superlattice.

In the revision, we have updated Supplementary Fig. 2 with a revised version (Fig. R6). Additionally, we have updated the statements in Lines 88-89 and included Fig. R7 as the new Supplementary Fig. 16, with relevant discussions provided in the new Supplementary Note 7. We have also added the reference mentioned by the Reviewer as ref. 33 in the revised manuscript.

Comment 3: *Lastly, the manuscript generally lacks detailed information about the experiments, such as EELS, which is crucial for validating the presented findings.*

Response: We thank the reviewer for his/her valuable comment. We conducted STEM-EDS to obtain atomic-scale elemental mapping. However, due to an oversight, we mistakenly referred to STEM-EDS as EELS in the original manuscript and supplementary information. In the revision, we have corrected the descriptions and included more detailed information about the experiments in the Methods section.

Detailed responses to reviewer #4:

General Comment: *Heterostructures and superlattices composed of layered transition metal dichalcogenides (TMDs) benefit from their superior emergent properties over individual components and provide significant promise for the development of multifunctional electronic devices. However, traditional synthesis methods for these structures depend on layer-by-layer artificial construction and are limited by their complexity and time-consuming. In this work, the authors develop a universal strategy for the automated synthesis of TMD superlattice single crystals through self-assembly, exemplified by the $NbSe_{2-x}Te_x$ 1T/1H superlattice, which retains the electronic characteristics of both 1T and 1H layers. This self-assembly for superlattice crystal synthesis approach could extend to other TMDs (such as $Nb_{1-x}(V/Ti)_xSe_2$) or layered materials, opening new avenues for efficient fabrication and broad applications of superlattices. Therefore, this simple method is helpful for the efficient development of new heterostructure superlattices and novel physical properties. It can be accepted after addressing the following comments:*

Response: We deeply appreciate the reviewer's positive assessment of our work. As shown in the response below, we have addressed the comment raised by the reviewer, and have revised the manuscript accordingly.

Comment 1: *Since the typical TMD $NbSe_2$ and $NbTe_2$ usually stabilize in the 2H type and 1T'' type, respectively, why does this $NbSe_{2-x}Te_x$ superlattice trend to stabilize in the 1T/1H superlattice, not 2H/1T'' or other superlattice? What is the principle factor for the stabilization of the 1T/1H superlattice?*

Response: We thank the reviewer for this comment. We respond to this comment in two aspects:

(1) For a given x in $NbSe_{2-x}Te_x$, the 1H and 1T layers usually exhibit very similar lattice constants. However, the 1T'' layer, which undergoes a 1×3 reconstruction, possesses a significantly larger lattice constant along the reconstruction direction compared to the corresponding reorganized lattice constant of the 1H or 1T layer. For example, the lattice mismatch between the 1H and 1T layers for $NbSe_2$ ($NbTe_2$) is 0.077% (0.38%), while the mismatch between the 1H and 1T'' layers is 1.6% (1.7%). Such larger mismatches between the 1H and 1T'' layers can lead to higher energies when forming superlattices. As a consequence, the $NbSe_{2-x}Te_x$ superlattice tends to stabilize in the 1T/1H superlattice rather than in the structures involving the 1T'' layers. Furthermore, we also examined the formation energies of the 2H/1T'' configuration in $NbSe_{2-x}Te_x$ with $x = 0.83$, as shown in Fig. R8. By considering dozens of random doping configurations, we obtained the formation energies, as illustrated in Fig. R9. These results confirm that the formation energy of the most stable 2H/1T'' configuration is

higher than that of the 1T/1H (6R) configuration.

Fig. R8 | Eight bulk configurations with different stacking orders of 1H or/and 1T layers for $\text{NbSe}_{2-x}\text{Te}_x$ with $x = 0.83$.

Fig. R9 | Formation energies of $\text{NbSe}_{2-x}\text{Te}_x$ ($x = 0.83$) with eight different stacking configurations.

(2) As discussed above, lattice mismatch is an important factor in preventing the presence of the 1T'' phase in superlattice structures. For the formation of the 1T and 1H superlattices, the close formation energies of the 1T and 1H phases play a pivotal role, facilitating the stable coexistence of these layers during growth. Furthermore, the stacking order and interlayer interactions contribute to fine tune the energies of various 1T and 1H superlattice configurations. For example, although the 4H_b, 4H_d, and 6R configurations all feature alternating 1T and 1H layers, $\text{NbSe}_{2-x}\text{Te}_x$ exhibits a preference for the 6R (1T/1H) structure. Additionally, charge transfer between neighboring 1T and 1H layers leads to electrostatic binding, further stabilizing the 1T/1H superlattice. These aspects have been thoroughly addressed throughout the paper.

In the revision, we have updated the statements in Lines 167-168, 170-171, and

included relevant discussions provided in the new Supplementary Note 9. We have also replaced Fig. 3d with Fig. R9 and replaced Supplementary Fig. 8 with Fig. R8.

Comment 2: *The authors claimed that some previous reports that adjusting the stoichiometric ratio of Te to Se can change the thermal stability of the T phase and H phase in ternary NbSe_{2-x}Te_x, such as the previous adjusting ratio of Te to Se in the NbSe_{2-x}Te_x to 1 could reach 1T-NbSeTe (e.g., J. Phys.: Condens. Matter 2020, 32 025702). From Figure 1a, it can be seen that 1T/1H NbSe_{2-x}Te_x 1T/1H superlattice was formed. Why does this new method cause the formation of NbSe_{2-x}Te_x 1T/1H superlattice when $x = 1$, not pure 1T type? It is better to give more explanation.*

Response: We thank the reviewer for this comment. We note that a similar issue was also raised by Reviewer #2 (Comment 2), with our detailed responses presented earlier and simply reproduced here. In Fig. 1a of the original manuscript, we confirm that NbSe_{2-x}Te_x single crystals form a 1T/1H superlattice at $0.64 \leq x \leq 0.89$ and a distorted 1T type at $x > 1.1$. To clarify the structure of NbSe_{2-x}Te_x at $x = 1$, we conducted STEM experiments on a NbSe_{2-x}Te_x single crystal with $x = 1.03$, showing a mixture of a distorted 1T type and a 1T/1H superlattice (see Fig. R7). These results suggest that, under our growth conditions, NbSe_{2-x}Te_x with $x \approx 1$ is in a transitional phase between the 1T/1H superlattice and the distorted 1T type. Given that different growth conditions often lead to structural variations for the transitional phase, it is reasonable that the previous work (J. Phys.: Condens. Matter 2020, 32 025702) reported NbSeTe as a 1T type at $x = 1$.

In the revision, we have updated the statements in Lines 88-89 and included Fig. R7 as the new Supplementary Fig. 16, with relevant discussions provided in the new Supplementary Note 7. We have also added the reference mentioned by the Reviewer as ref. 33 in the revised manuscript.

Comment 3: *Figure 1d shows three types of crystal structures in these ternary NbSe_{2-x}Te_x compositions, which are distinguished by color regions. Is it curious there are no mixing phases between these regions?*

Response: We thank the reviewer for this insightful comment, which also relates to Comment 2. Following the reviewer's suggestion, we identify mixed phases in the NbSe_{2-x}Te_x single crystal at $x \approx 1$, as shown in Fig. R7. As the reviewer pointed out, mixed phases are indeed present in the transitional region ($0.89 < x < 1.1$).

In the revision, we have updated the statements in Lines 88-89 and included Fig. R7 as the new Supplementary Fig. 16, with relevant discussions provided in the new Supplementary Note 7.

Comment 4: *Are there unique physical properties, such as superconductivity, for this 1T/1H NbSe_{2-x}Te_x 1T/1H superlattice?*

Response: We thank the reviewer for his/her comment. Following the reviewer's suggestion, we conducted low-temperature transport experiments on $\text{NbSe}_{2-x}\text{Te}_x$ single crystals. As shown in Fig. R10, the $\text{NbSe}_{2-x}\text{Te}_x$ 1T/1H superlattice exhibits superconductivity, with a transition temperature between 1.6 and 2.5 K, positioned between NbSe_2 and NbTe_2 . This study mainly focuses on the general synthesis strategy of the 1T/1H superlattice. We intend to explore the properties of the 1T/1H superlattice, including its superconducting behavior, in future research.

In the revision, we have incorporated Fig. R10 as the new Supplementary Fig. 17, with relevant discussions as the new Supplementary Note 8.

Fig. R10 | Superconductivity of $\text{NbSe}_{2-x}\text{Te}_x$ single crystals. **a** Normalized resistance (R_N representing the normal-state resistance) as a function of temperature for both in-plane (blue line) and out-of-plane (red line) measurements. These results indicate a superconducting transition at 1.6 K in the 1T/1H superlattice ($x = 0.89$). The superconducting transition temperatures of the 1T/1H superlattice are nearly identical along different crystal orientations. **b** Structural phase and superconducting transition temperature as functions of doping level x in $\text{NbSe}_{2-x}\text{Te}_x$.

Responses to the reviewers
(Manuscript NCOMMS-24-45261A)

Detailed responses to reviewer #1:

General Comment: *I am satisfied with the revisions made in the latest version.*

Response: We appreciate the reviewer's positive feedback on the latest version.

Detailed responses to reviewer #2:

General Comment: *I co-reviewed this manuscript with one of the reviewers who provided the listed reports.*

Response: We thank the reviewer for dedicating his/her time and effort to co-review our manuscript.

Detailed responses to reviewer #3:

General Comment: *The authors have adequately addressed the concerns initially raised. Given that the self-assembly of the 1T/1H superlattice is primarily influenced by the equilibrium between the formation energies of the various phases, it would be beneficial to include a more comprehensive introduction, properly citing the two papers that were discussed in the initial review on the stacking of 1H, 2H, and 3R types. Once these revisions have been made, I am satisfied with publishing this work.*

Response: We appreciate the reviewer's valuable comments. We have revised the introduction accordingly and added the references mentioned by the reviewer as ref. 32 and ref. 33 in the revised manuscript.

Detailed responses to reviewer #4:

General Comment: *The authors have addressed all my questions, it can be accepted now.*

Response: We appreciate the reviewer's recognition of our previous responses and thank them for recommending our manuscript for publication.

The authors present a universal approach for the automated synthesis of TMD superlattice single crystals via self-assembly, demonstrated using the NbSe₂-xTex 1T/1H superlattice. The central concept of this method lies in balancing the formation energies of the T (octahedral) and H (trigonal prismatic) phases. By fine-tuning the Te to Se stoichiometric ratio in NbSe₂-xTex, the authors successfully minimize the formation energy difference between the T and H phases, thereby facilitating the self-assembly of 1T and 1H layers into a coherent 1T/1H superlattice. The authors initially focused their study on Te-doped NbSe₂ and subsequently extended their investigation to include V-doped and Ti-doped NbSe₂ systems.

One technique concern lies with the X-ray diffractometer technique employed for structural determination. The authors have described the instrument as a single crystal X-ray diffractometer. However, in the experimental section, they state, 'XRD patterns were collected using a Rigaku SmartLab SE X-ray diffractometer with Cu K α radiation ($\lambda = 0.15418$ nm) at room temperature.' It is important to note that the Rigaku SmartLab SE is a powder X-ray diffractometer. Therefore, I recommend that the authors revise their description to accurately reflect this by stating that a 'powder X-ray diffractometer was applied to single crystal samples.'

Let's closely examine the structural 'models' presented. I refer to them as 'models' because it is unclear how the authors determined the precise atomic coordinates using only powder X-ray diffraction and STEM techniques. The figure below corresponds to Fig. 1b in the main text. The structural differences between the 2H and 3R phases can be simplified as the packing of Nb@(Se/Te)₆ trigonal prisms or octahedra. In the 2H phase, the Nb@(Se/Te)₆ trigonal prisms are packed in alternating orientations, whereas in the 3R phase, the Nb@(Se/Te)₆ trigonal prism layers are interspersed with layers of octahedral Nb@(Se/Te)₆. My first question concerns the Se/Te atomic mixture. How did the authors determine the atomic occupancy for each site? For instance, in the 3R phase, the authors labeled the octahedral layers as containing only Nb@Se₆. What method was used to determine this—STEM, perhaps? If it is true that Se fully occupies the octahedral layers, an inconsistency arises: how can the lattice parameters remain constant as the Te doping amount increases in the 1T/1H phases?

Let's now examine the STEM image presented in Fig. 2a. Could the authors specify the metal-to-metal atomic distances, specifically the Nb-Nb distance, in the 1H and 1T phases separately? Additionally, it appears that the electron distribution in the 1T layer shows a dumbbell-like shape, I marked in red in the figure below. Could the authors provide more insight or confirmation regarding this observation?

The energy difference shown in Fig. 3 between the 1T/1H and distorted 1T phases is quite small. In fact, the distorted 1T phase exhibits a lower energy than the 1T/1H phase at around $x(\text{Te}) \sim 0.8$. Given this, how can the authors assert that the formation energy difference between the T and H phases is the primary factor controlling phase stability?

In Fig. 4, for the V-doped system, the metal atoms at the 1H and 1H' sites appear brighter than those at the 1T and 1T' sites. Did the authors verify which sites have a higher concentration of V atoms? Additionally, this phenomenon is not as clearly observed in the Ti-doped system. Why was the doping concentration of V set at 0.1 and Ti at 0.05? Was there a specific reason for not using the same doping levels for V and Ti?

In summary, the authors should address and clarify the questions listed above.